# Inspiration for Seismic Diffraction Modelling, Separation, and Velocity in Depth Imaging

**Yasir Bashir** [1,2,*] **, Nordiana Mohd Muztaza** [1] **, Seyed Yaser Moussavi Alashloo** [3] **,**
**Syed Haroon Ali** [4] **and Deva Prasad Ghosh** [2]

1   School of Physics, Universiti Sains Malaysia, USM, Penang 11800, Malaysia; mmnordiana@usm.my
2   Centre for Seismic Imaging (CSI), Department of Geosciences, Universiti Teknologi PETRONAS,
    Seri Iskandar 32610, Malaysia; ghosh_deva@yahoo.com
3   Institute of Geophysics, Polish Academy of Sciences, 01-452 Warsaw, Poland; y.alashloo@gmail.com
4   Department of Earth Sciences, University of Sargodha, Sargodha 40100, Pakistan; geologyuos@gmail.com
*   Correspondence: Yasir.bashir@usm.my or dryasir.bashir@live.com; Tel.: +60-16-2504514

**Abstract:** Fractured imaging is an important target for oil and gas exploration, as these images are heterogeneous and have contain low-impedance contrast, which indicate the complexity in a geological structure. These small-scale discontinuities, such as fractures and faults, present themselves in seismic data in the form of diffracted waves. Generally, seismic data contain both reflected and diffracted events because of the physical phenomena in the subsurface and due to the recording system. Seismic diffractions are produced once the acoustic impedance contrast appears, including faults, fractures, channels, rough edges of structures, and karst sections. In this study, a double square root (DSR) equation is used for modeling of the diffraction hyperbola with different velocities and depths of point diffraction to elaborate the diffraction hyperbolic pattern. Further, we study the diffraction separation methods and the effects of the velocity analysis methods (semblance vs. hybrid travel time) for velocity model building for imaging. As a proof of concept, we apply our research work on a steep dipping fault model, which demonstrates the possibility of separating seismic diffractions using dip frequency filtering (DFF) in the frequency–wavenumber (F-K) domain. The imaging is performed using two different velocity models, namely the semblance and hybrid travel time (HTT) analysis methods. The HTT method provides the optimum results for imaging of complex structures and imaging below shadow zones.

**Keywords:** diffraction modelling; double square root equation; travel time; dip frequency filtering; migration

## 1. Introduction

Seismic diffraction events are produced because of small-scale elements in the subsurface, such as faults, fractures, channels, and rough edges of salt bodies; or because of small changes in the seismic reflectivity, such as those produced by fluid occurrence or fluid movement in the period of production. These diffracted waves contain the most important information for subsurface discontinuities [1]. The diffraction theory sometimes makes it impossible to understand the qualitative properties of diffraction phenomena [2]. Further, these diffraction data are also used for travel time approximation for anisotropic imaging [3]. The separation of diffraction is widely used [4–7] for enhancement of seismic imaging, such as fault, fracture, complex variable velocity structure, and karstification imaging [8]. Further, the diffraction response for a nonzero separation of the source and receiver was presented by Berryhill in 1977 [9], in which theoretical support was obtained for applying the zero-separation theory to stack the seismic data. Amplitude preservation of seismic waves was achieved by Hilterman in

1975 [10] by using the front sweep velocity approach, in which a graphical method evolves. The last decade was devoted to theoretical work and many methods were introduced, however the lack of implementation in the real world was limited due to computing power and facilities. In this decade, the computation increased drastically and the technology development increased compared to the last decade. Diffraction enhancement in prestack seismic data was introduced by Bansal in 2005 [11]. He introduced the most effective techniques, which involve the decomposition of seismic gather data into eigensections and flows based on radon transformations. Diffraction imaging using the multifocusing method was used in 2009 by Berkovitch [12], in which the diffraction focusing stack (DMFS) was used to separate the diffraction energy in a stack section and defocus the reflection energy over a large area. Diffraction separation has been widely used by Fomel since 2002 [13] using the Claerbout principle [14], and has contributed to the field of seismic diffraction imaging.

Diffraction hyperbolic patterns occur frequently in seismic sections, and their existence is usually taken as evidence of abrupt discontinuities in the subsurface reflector geometry [10]. Geophysicists understand ray path geometry; for every "point" diffractor, the edge of the reflector and fault give rise to a hyperbolic pattern in a zero-offset section [9]. In an inclined case, a series of diffractions are originated along the fault surface so that the apex of each curve is on the fault plane.

A conventional processing workflow suppresses the diffractions and enhances the reflections. Diffracted and reflected seismic waves are fundamentally different physical phenomena [15]. As shown in Figure 1, the source and receiver are on the surface. Once they are active, the reflections are recorded on a continuous reflector and a diffraction hyperbola is produced at the edges. This is because of the lateral acoustic impendence contrast. Further diffraction hyperbolas have positive to negative amplitudes, which generally decrease with time. The common practice during processing is that reflected waves are tuned and diffracted waves are suppressed by being considered as noise, but these diffracted events carry the most important information about the subsurface.

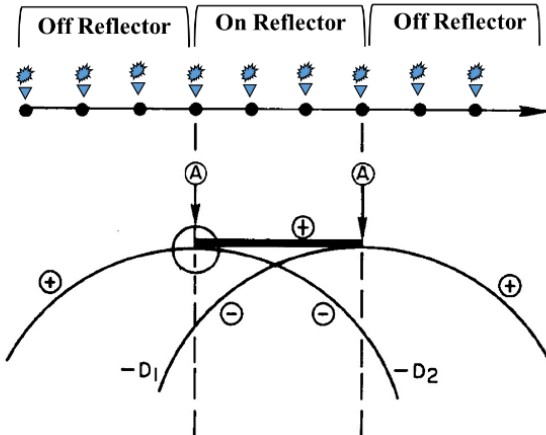

**Figure 1.** Graphical representation of the diffraction phenomenon with the geometry of the source and receiver functions produced at the edges of the reflectors (D1 and D2). A phase change of 180° is shown on either side of the curves [16].

The separation of seismic diffraction is not an innovative idea. A previous study [17] used diffraction to estimate the velocity from forward modeling and the local slant stack. Another study [18] used a common diffraction point section for imaging of diffraction energy and detection of local heterogeneities, while another study [19] simulated diffraction responses to enhance the velocity analysis.

In this paper, we have developed a unique approach by integrating the previous solutions used for multiple eliminations. This method can separate diffraction energy and suppress the reflection amplitude, which is necessary for Amplitude versus offset (AVO) analysis [20]. For imaging purposes, a simple and fast method is used, which follows the one-way wave equation migration, as explained

below. We started with the zero-offset or processed stacked seismic data as the input and applied the Fourier transform to convert our data from the time domain to the frequency domain.

## 2. Materials and Methods

### 2.1. Diffraction Modelling

The principle of the diffraction theory is mainly based on the subsurface behaving similarly to acoustic media. This means that shear waves are not considered for diffraction studies; only primary waves cause the production of diffraction with low reflectivity and constant velocity. The scalar wave equation is used to satisfy the wave equation. The reflector is an ensemble of individual elements and the response is the sum or integration of each element. The zero-offset solution is available and further planning to extended for the nonzero-offset.

When the source and receiver are separated, the envelope of arrival times (diffraction curve) has a different shape due to the increased travel times. This can be explained by the "double square root equation (DSR)" for travel time, which is:

$$T_L = \sqrt{\left(\frac{Tm}{2}\right)^2 + \left[\frac{L - \frac{x}{2}}{V}\right]^2} + \sqrt{\left(\frac{Tm}{2}\right)^2 + \left[\frac{L + \frac{x}{2}}{V}\right]^2} \tag{1}$$

where *Tm* is the time, *V* is the velocity, and *x* is the separation between the normal incident (single square root equation) and the separated shot receiver (double square root equation). The curve is greatest at the observation location over the point source. At location *L*, the velocity becomes smooth and Equation (1) simplifies into:

$$T_L = \sqrt{Tm^2 + \frac{L^2}{V^2}} \tag{2}$$

The above is the usual expression used for reflection times from a horizontal reflector. This observation forms the basis of the velocity determination method.

Using Equation (2) in MATLAB, we observed the behavior of diffraction hyperbola by taking different background velocity values, such as 2000, 3500, and 5000 m/sec, as shown in Figure 2.

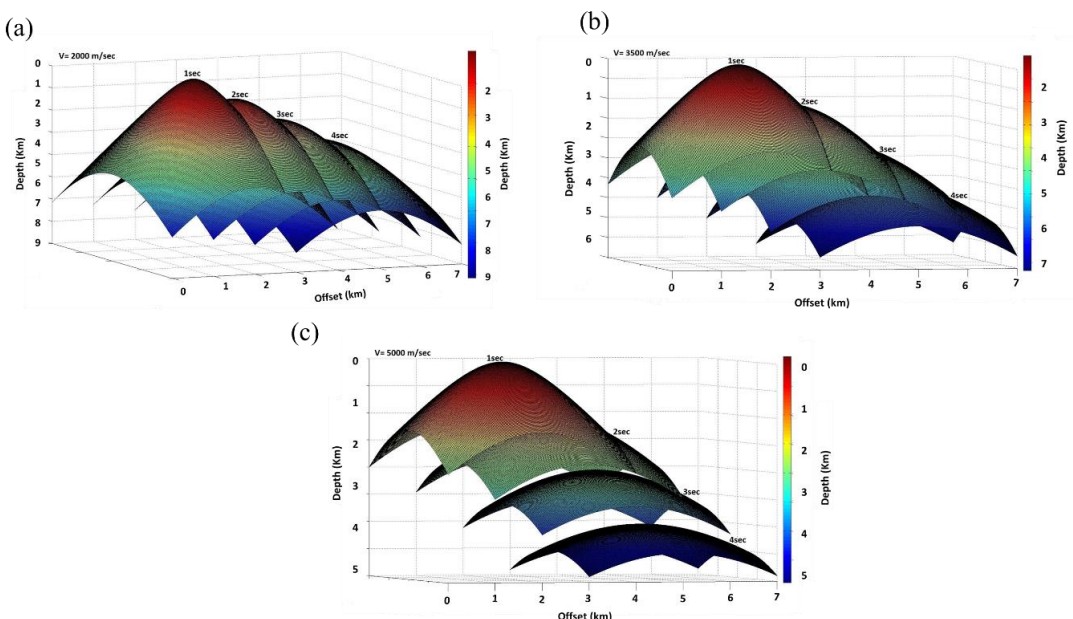

**Figure 2.** Diffraction hyperboloids with constant velocity at: (**a**) 2000 m/sec, (**b**) 3500 m/sec, and (**c**) 5000 m/sec. The curvature of the hyperboloids is spread out with an increase of velocity.

We model the 3D diffraction curves on a point diffractor of different velocities. Figure 2 explains the hyperbolic pattern of diffraction concerning offset and depth; as the depth increases, the curvature of the hyperbola is smothered. The effect of velocity can be found by comparing Figure 2a–c. The scale of all parts of the hyperbolas is the same on the *X*-axis (offset) and *Y*-axis (depth), but there is a change of the velocity in the point diffractor. Velocity directly affects the diffraction hyperbolic curvature, such as in a high-velocity area. The diffraction pattern is more complex and more difficult to interoperate during processing. Separation of diffraction events, which are complex in deeper regions, will be more difficult because of their similarity with reflection data.

Further, we extend our algorithm from constant velocity to variable velocity. Increasing and decreasing velocity diffraction behavior can be observed in Figure 3a,b, respectively. The shape of the diffraction curve at 1000 m/sec velocity changes as the depth changes. This analysis indicates that the diffraction curve is not only dependent on velocity but also on depth.

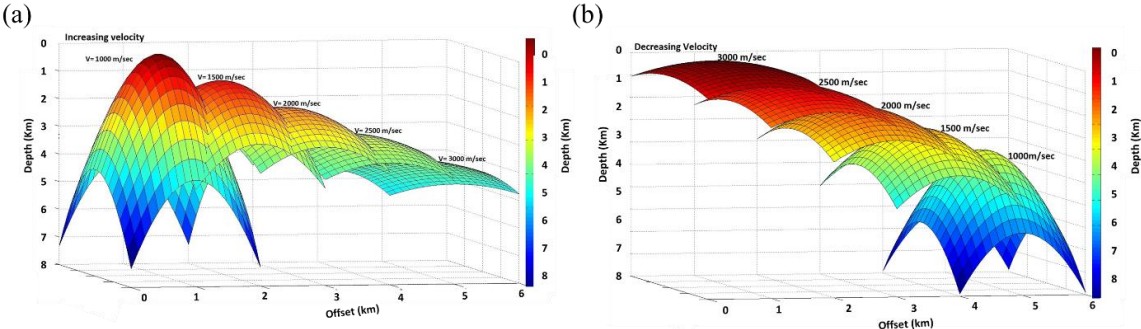

**Figure 3.** Diffraction hyperboloids: (**a**) increasing velocity with depth and (**b**) decreasing velocity with depth.

## 2.2. Dip Frequency Filtering

Filtering is a well-known method used to separate seismic events that are available in the seismic data, using the suitable apparent velocity as the selection criterion. The gradient of apparent velocity, $V_{app} = dx/dt$, will give the arrivals for the seismic data section.

The one-dimensional Fourier transform $F(u)$ of a single variable, namely a continuous function $f(x)$, is defined by the following equation [21]:

$$F(u) = \int_{-\infty}^{\infty} f(x) \exp\left(-j2\pi ux\right) dx \tag{3}$$

where $= \sqrt{-1}$. Conversely, given $F(u)$, we can obtain $f(x)$ by employing the inverse Fourier transform, as given below:

$$f(x) = \int_{-\infty}^{\infty} F(u) \exp\left(j2\pi ux\right) \tag{4}$$

These two equations contain the pair of Fourier transforms and were used to convert our data from the time domain to the frequency domain. They were easily extended to the two variables $u$ and $v$.

For the illumination of the coherent wavefront of the constant amplitude, the distribution of the amplitude in the spatial frequency spectrum $b(x, y)$ and the object plane $B(u, v)$ can be defined by using the inverse Fourier transform [22]:

$$b(x, y) = \int_{-\infty}^{\infty} \cdot \int_{-\infty}^{\infty} B(u, v) \exp\left(-i2\pi(ux + vy)\right) du\, dv \tag{5}$$

Alternatively, it can be explained using a simplified notation, where $\rightleftharpoons$ shows the pair of Fourier transforms.

$$b(x, y) \rightleftharpoons B(u, v) \tag{6}$$

A spatial frequency spectrum is defined as the description of the image forming process.

$$B'(u'v') \rightleftharpoons b(x, y). \, a(x, y) \tag{7}$$

$$= B'(u', v') \, {}^* A\,(u'v') \tag{8}$$

where $A\,(u'v') \rightleftharpoons a(x, y)$ is called the spread function, the * symbol shows the convolution, and $(u', v')$ are the coordinates of the spectrum window for magnification of the filter amplitude.

Filtering is applied to the frequency–wavenumber (F-K) spectrum to separate the diffracted events from the reflection data by defining a filter, in which slopes monotonically increase and amplitudes of the data correspond to the slopes of dipping events, for example in an inclined structure:

$$Slope = \frac{dt}{dx} = \frac{1}{V_{app}} \tag{9}$$

### 2.3. Diffraction Separation

The underlying hypothesis works for modeling and separating the diffracted events from the reflection seismogram. In zero-offset or stack data, the reflection events have a higher and strong amplitude. Removing those events exposes other comprehensible information, often in the form of seismic diffraction. We applied the Fourier transform and converted our data from the time domain to the frequency domain. The F-K spectrum is the sketch of the amplitude distribution with the wave number. Through minimizing the forecast for outstanding work, many iterative tasks were performed whilst developing the local slope filter to remove the specular reflection from the seismic data. An equivalent impression with an application based on the prediction error filters was investigated by Claerbout (1994). Even though separation of reflection and diffraction energy is never precise, our procedure satisfies the practical purpose of improving the wave response of small subsurface discontinuity, such as for faults, fractures, Karst sections, and unconformities. The obtained research results show that the goals of filtering and separating the diffraction from full-wave seismic data were achieved.

### 2.4. Imaging Diffraction

The various imaging algorithms used for the wave equation revealed similar structural imaging; however, the same processing capabilities and faster implementation were considered using these techniques. Here, we used a constant velocity earth model, while for imaging purposes the Stolt migration method was used. Corresponding to other migration methods, this migration method can be retreated and made into a modeling program. One of the drawbacks or limitations in the principle is that the Stolt migration is imitated in the depth variation of the velocity for better imaging. The simplest sketch of the Stolt migration method can be defined by the following equations after applying the idea that the migrated image at the location of $(x, z)$ is the exploding reflector wave at time $t = 0$:

$$k_z = \mp \sqrt{\frac{w^2}{v^2} - k_x^2} \tag{10}$$

$$P(k_x, k_z, t = 0) = \left[\frac{v}{2} \frac{k_z}{\sqrt{k_x^2 + k_z^2}}\right] P\left[k_x, 0, \omega = \frac{v}{2}\sqrt{k_y^2 + k_z^2}\right] \tag{11}$$

where $P(k_x, z = 0, \omega)$ is the zero-offset section and $P(k_x, k_z, t = 0)$ is the migrated section in the frequency–wavenumber domain.

*2.5. Regional Geology*

In the southeast Asian basins, there are several geophysical challenges [23]. Some of them are listed below:

- Imaging thin sands, which is often beyond seismic resolution;
- Imaging below gas clouds;
- Karst imaging in carbonate fields;
- Fractured basement imaging and its internal architecture;
- Understanding the wave propagation of ineffective media and related anisotropy;
- Velocity analysis and anisotropy;
- Multiple eliminations;
- Diffraction imaging for small-scale events.

The focus of this research is to image the fractures that are the main challenges faced in the field, including fractured basements, fracture distribution, connectivity, and lateral variation, which cause poor seismic imaging.

A regional map of the Malaysian basin is shown in Figure 4a—on the left side is the Malay Basin, the right side shows the Sabah Basin, and the below-right side shows the Sarawak Basin.

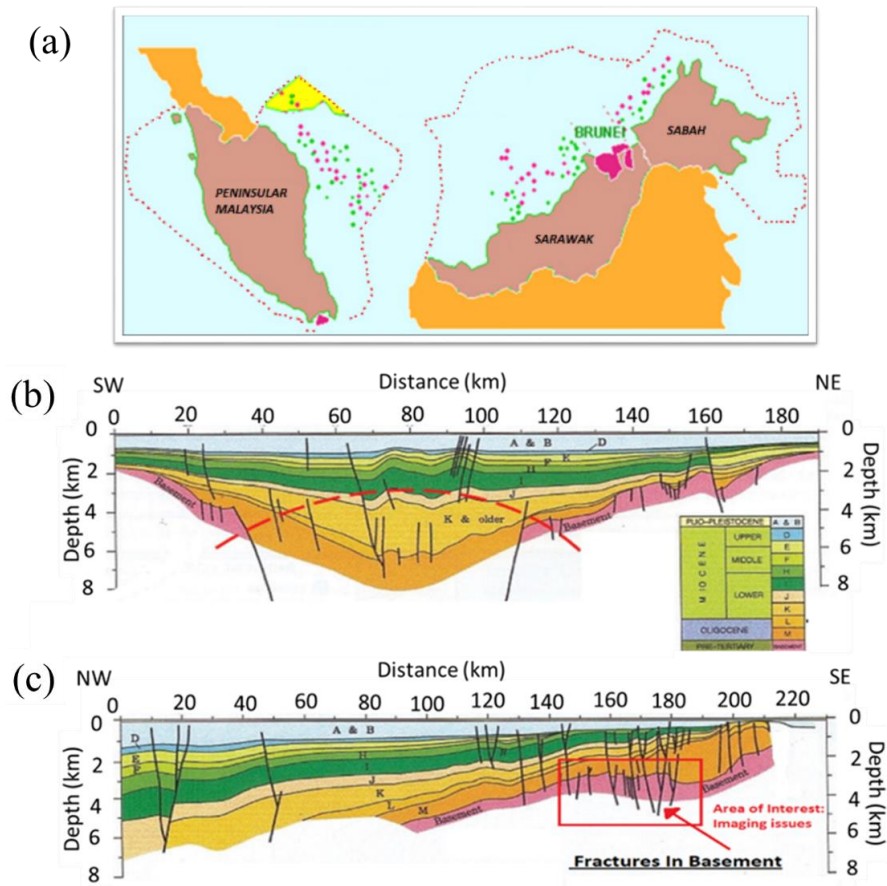

**Figure 4.** (**a**) Geographical map of the Malay Basin. Shown are the provinces of Peninsular in the Malay Basin, Sarawak, and Sabah Basin [24]. The Malay Basin subsurface structure shown with a regional cross-section view. (**b**) South-west to north-south section. The fractured basement can be seen at a depth of 2–5 km and varies with lateral extension. (**c**) North-west to south-east section [5].

As mentioned, our objectives are to focus on fractures, so in this study the Malay Basin is considered. A regional cross-section of the Malay Basin is shown in Figure 4b,c. As already mentioned, the difference between a good well and a dry well is whether it encounters main fracture corridors, which must be known before drilling. However, the identification of fractures is impossible without imaging the fractured basement, and these complex structures cannot be identified by conventional imaging alone.

## 3. Results and Discussion

### 3.1. Detection of Faults and Fractures

In general geology, a fracture is any kind of separation or break in a rock formation. Examples are joints or faults in which the formation is divided into two or more pieces [25]. In geology, fracture is a broad term that includes faults, fractures, and discontinuities. These fractures can provide access for fluids, such as water or hydrocarbons, to move into the rocks [26]. Therefore, the model shown in Figure 5a is the fractured basement model from the Malay Basin. The number of faults and fractures present in the igneous basement are a challenge for seismic imaging. This field produces oil from the fractured basement. For simplicity in testing the diffraction imaging algorithm and concept, we extracted a part of the fractured model, which is shown in Figure 5b.

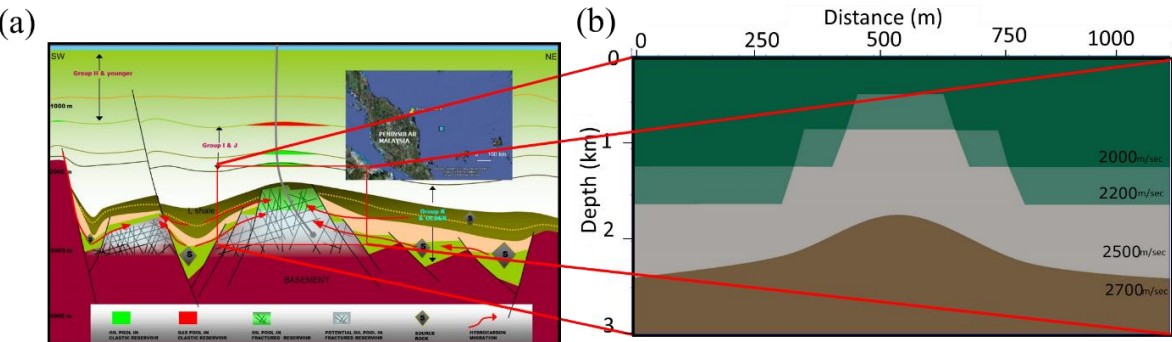

**Figure 5.** (**a**) Malay Basin field, which contains extensive faults and fractures in the igneous basement [28]. (**b**) Input initial velocity model for seismic analysis [8].

A finite difference modeling technique is performed to generate the gather data using a zero-offset configuration. Figure 6a shows the zero-offset section; diffractions are produced at the edges of the reflector and a series of diffraction curves appear on the fault plane. The amplitude of the hyperbola decreases with increasing depth because of the attenuation factor. Furthermore, the curvature of the hyperbola is higher in the shallow section than in the deeper section. A 3D projection of the hyperbolic curves is shown in Figures 2 and 3, which describe the fundamentals of the diffraction with respect to velocity and depth.

The frequency–wavenumber (F-K) analysis is a standard array technique that simultaneously calculates the power distributed among different speeds and directions of approach [27]. Figure 6b shows the results in the F-K domain for the zero-offset data. Here, we can see that the amplitude from the reflection occurs at a speed of zero cycles per second. Further, the diffraction amplitude is shown away from zero with the lower amplitude. The filter, which is designed based on the dip of the frequency cut-off, removes the reflection and preserves the diffraction only by looking into the spectrum. This filter was design based on trial and error analysis. Finally, the output of the filter is shown in Figure 6c. Figure 6d shows the diffraction section, which is achieved by DFF. The reflection is removed and diffraction is enhanced. A series of diffractions appears on the fault plane. Diffraction stacking can help in fault interpretation.

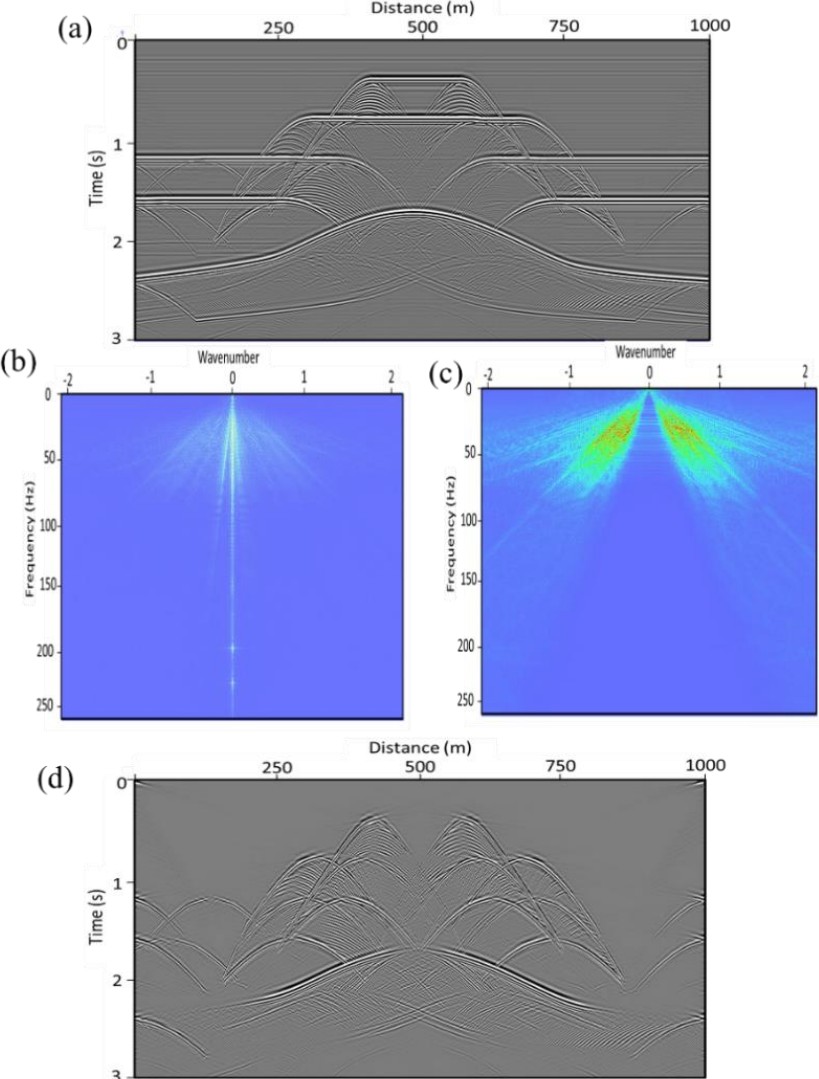

**Figure 6.** (**a**) Seismic response of the input velocity model; a series of diffractions are generated at the fault location. (**b**) The frequency–wavenumber (F-K) spectrum, which shows the amplitude distribution with wave cycles per kilometer. (**c**) F-K spectrum after application of the designed filter; only the diffraction amplitude is displayed, while the reflection amplitude has been removed. (**d**) After separation using the DFF filtering algorithm, diffractions are preserved and reflections are successfully suppressed.

Diffraction stacking or imaging can add information to an interpreter for fault and fracture identification. The Stolt migration is applied to the diffraction and reflection data, which are shown in Figure 7a,b. The diffractions are imaged properly and the point diffractions show the locations of the faults and a probable discontinuous surface. Figure 7b shows the migrated synthetic seismic data without diffraction separation; not all of the components of the faults are illuminated, as desired. Furthermore, separating the diffractions would help us to interpret the faults, as the apex of each diffraction shows the continuity of the fault plane. Figure 7c is the amplitude spectrum of the migrated data in the time–distance (t-x) domain, which shows the amplitude distribution with the trace number. At the fault location, the amplitude is very low or near to zero because most of the migration algorithms kill the diffractions that are considered as noise. In this context, diffraction separation is needed to improve the imaging results.

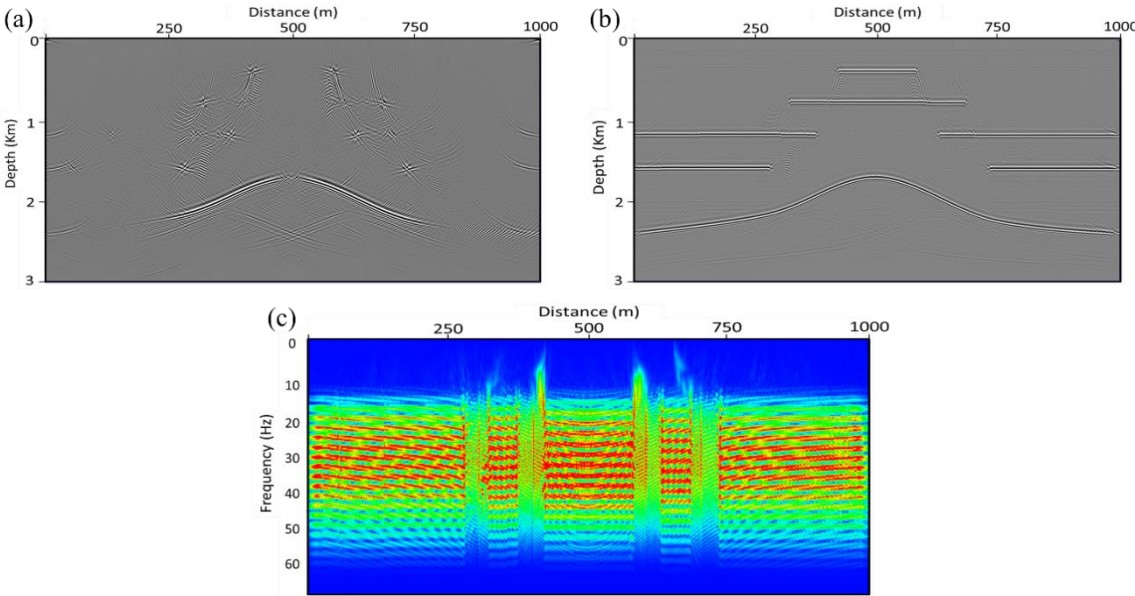

**Figure 7.** (**a**) Migrated seismic section, The point diffractor shows the location of the discontinuity and the fault indicator. (**b**) Full-wave seismic migrated section; all components of the fault are missing. (**c**) Amplitude spectrum (frequency bandwidth) of the migrated data, showing that the diffraction amplitude is biased.

### 3.2. Generalized Workflow

This work presents a simple and robust approach based on a frequency domain algorithm along with a user-friendly interface. Usually, in a stack data, the reflection events have higher and stronger amplitudes. Removing those events exposes other comprehensible information, often in the form of seismic diffraction. After using a Fourier transform to convert our data from the time domain into the frequency domain, the F-K (frequency–wavenumber) spectrum shows the amplitude distribution with the wavenumber. Through minimizing the forecast for outstanding work, many iteration tasks were performed whilst developing the local slope filter for separation of specular reflection from the full-wave seismic behavior, which is based on equivalent impression and prediction error filters [5,29]. The separation of reflection and diffraction energy is accurate, as this method accommodates wave responses from small subsurface discontinuities, such as faults, fractures, karst sections, and unconformities, which appear as diffraction in the data. The uniqueness of our methodology is the integration of essential approaches for diffraction modeling, separation, and imaging. A generalized workflow and steps taken to perform diffraction separation for data are given below.

❖ Generate a velocity model, i.e., the fractured or Marmousi model;
❖ Use finite difference wave propagation to record the seismic behavior in the provided geological model;
❖ In the case of raw data from the field, initial processing is needed to improve the data quality (including sorting, CDP ordering, NMO correction, and stacking, if the data are prestacked);
❖ Apply a Fourier transform to observe the spectrum of amplitude distribution in the frequency wavenumber;
❖ Design the filter based on the dip component of the data in the dt/dx plane;
❖ Convolute the proposed dip frequency filter (DFF) with seismic data to separate the diffraction from reflection;
❖ Apply an inverse Fourier transform to convert the frequency–wavenumber data into the time–distance domain.

### 3.3. Marmousi Model

The Marmousi model was generated at the Institute Francais du Petrole (IFP) and is used to test imaging tools to improve and enhance imaging-related algorithms. The original Marmousi data set is generated using a 2-D acoustic finite difference modeling program [8].

The seismic imaging target zone is a reservoir located at a depth of about 2500 m. The model contains many reflectors, steep dips, and strong velocity variations in both the lateral and vertical directions (with a minimum velocity of 1500 m/s and a maximum velocity of 5500 m/s).

Figure 8a shows the Marmousi model in the (*x-z*) domain, which contains a huge number of faults and folds, representing a challenging distribution of velocity anomalies and discontinuities. The configuration of the survey was set up as an end-on shooting geometry, and the data were recorded in the shot domain, as shown in Figure 8b. Extensive processing for improving the signal to noise ratio and suppressing dispersion artifacts is applied to the raw data for diffraction enhancement. A close-up of data for the three shots is shown in Figure 9 before and after normal moveout (NMO) correction, which shows a flatness of moveout in the gathered data.

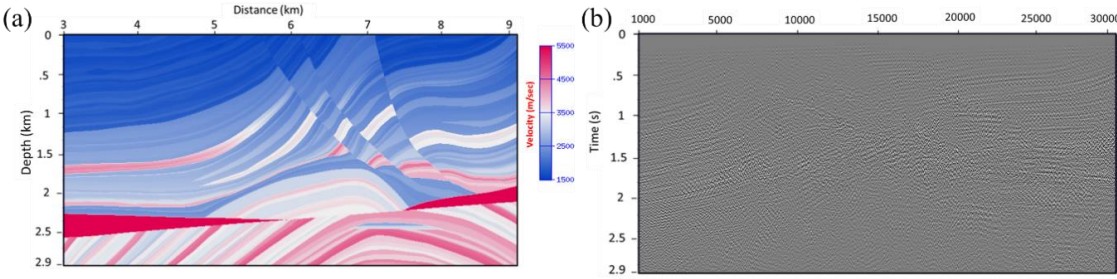

**Figure 8.** (**a**) Marmousi velocity model with a velocity range from 1650 to 4600 m/sec. Three major faults containing unconformity and overburden cause complexity in the velocity model for anticline structures. (**b**) Synthetic shot gather data are generated using finite difference modeling.

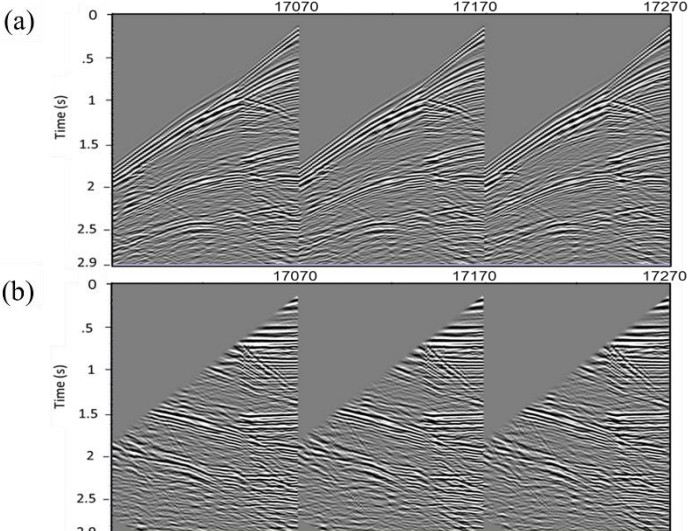

**Figure 9.** Seismic gather data (**a**) before the NMO correction and (**b**) after the NMO correction. Events are slightly straight at the true reflection points.

### 3.4. Importance of Velocity for Seismic Depth Imaging

Velocity plays a key role in various aspects of seismic methods, especially in migration, as a correct velocity is needed for proper stacking of the diffracted energy from the flanks of the hyperbola to the apex, as well as for the reflection. A velocity field is essential for field development planes, including but not limited to:

(a)　Gain derivation of wavefield divergence and attenuation;

(b)　Multiple eliminations, wavefield continuation, or Radon demultiple processes;

(c)　Seismic imaging for structures and reservoirs;

(d)　Lithology prediction through Vp/Vs (Velocity of P-wave and S-wave) and elastic inversion;

(e)　Pressure prediction;

(f)　Porosity prediction from acoustic impedance (AI);

(g)　Time-epth (T-X) conversion;

(h)　Volumetric calculation;

(i)　Net sand map plots.

Velocity estimation is a key processing parameter. The sensitivity and accuracy can be significantly improved with the use of a long spread survey design and by consuming higher-order terms. An anisotropic model may be required in certain geologic situations.

To enhance the processing and imaging accuracy, a velocity analysis was performed using two different methods:

(1)　Semblance analysis;

(2)　Hybrid travel time.

Semblance analysis is the conventional way of calculating velocity during the NMO correction. It allows for the refinement of seismic data. This is done by developing a velocity spectrum display to determine the velocity through different layers at depth. The velocity data are used to correct the curves of the hyperbolas and create a flat line, where all points are at an equal depth.

The hybrid travel time method is described in Figure 10, which shows the raytracing phenomena in a complex and heterogeneous medium with 100 rays. Paraxial optics can be defined as ray-tracing accomplished within the limits of very small ray angles and heights. This allows us to make a number of simplified assumptions that make the raytracing arithmetic considerably easier.

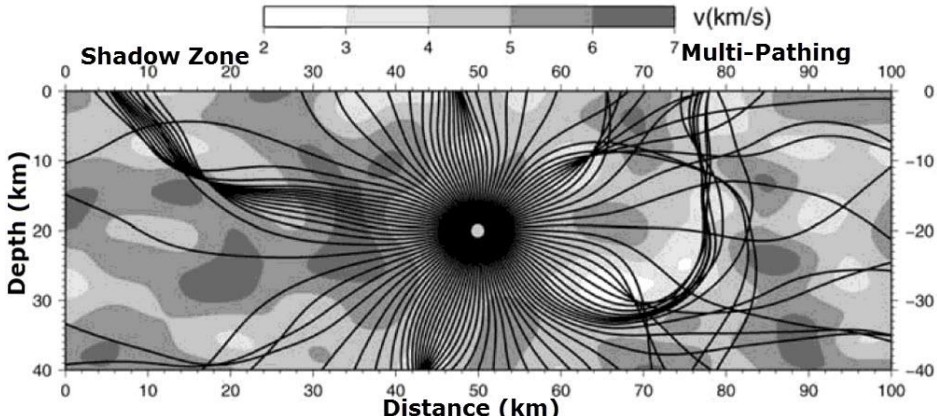

**Figure 10.** Trajectories of 100 rays emitted by a source point in a heterogeneous medium. Shadow zone and multipathing effects can be seen [30].

The eikonal differential equation is used for the elementary mathematical models, which describe the travel time (eikonal) propagation in a velocity model. The benefits of this method in comparison with ray tracing techniques are the capability to work in consistent model grids, the wide-ranging coverage of the receiver space, and the reasonable numerical robustness. A common implementation of the finite-difference eikonal equation is calculating the first-arrival travel times though frequency-dependent enrichment.

The eikonal equation, describing the travel time propagation in an isotropic medium, has the following form:

$$(\nabla \tau)^2 = n^2(x, y, z) \tag{12}$$

where $\tau(x, y, z)$ is the travel time from the source to the receiver point with coordinates $(x,y,z)$ and n is the slowness at that point (the velocity v equal to 1/n) [31].

This method calculates each travel time table using paraxial raytracing, then in the shadow zone (Figure 10) the travel times are computed by solving the eikonal equation.

### 3.5. Diffraction Separation and Imaging using Correct Velocity Model

After careful consideration of the velocity analysis and stacking process (Figure 11a), the data are ready for DFF as an input. Figure 11b shows the separated diffraction with an enhanced amplitude, which allows the hyperbola to be observed more accurately. For quality control (QC) purposes, full-wave and diffraction difference sections are generated to ensure the accuracy of the algorithm, as shown in Figure 11c.

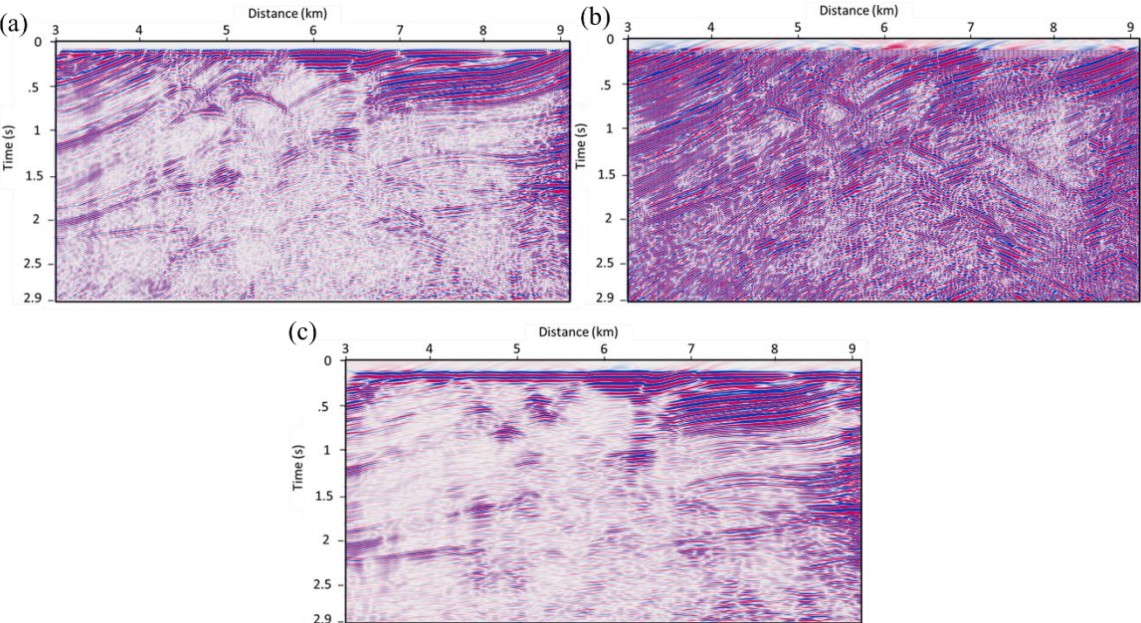

**Figure 11.** (**a**) Common Mid Point (CMP) stacked section after careful processing, including sorting of the gather data from the CMP gather data, amplitude correction, and NMO correction. (**b**) Diffraction section after application of dip frequency filtering with high amplitude display. Diffracted events are enhanced. (**c**) Difference or reflection section, showing that consideration of the diffraction is quite important for successful subsurface imaging, especially for faults and fractures.

We considered two methods to compare the results of imaging using different velocity modeling techniques. Figure 12a shows the image section using semblance velocity analysis, which was used because of uncertainties in picking the velocity, as this is an interactive process and the chance of error is greater. In this analysis, the three major faults are not imaged and interpretation is difficult. Further, the aim of migration was to delineate the reservoir below 2.5 km, which was not resolved because of velocity error.

The second method used for migration is the hybrid travel time method, which incorporates both the eikonal solver and the paraxial raytracing algorithm to calculate the velocity. It also handles the shadow zone and multipathing effects in the data. Figure 12b is a migrated seismic section with an accurate velocity model, which enhances the resolution as well as the structural clarifications. Further, the three major faults are interpreted accurately, as the resolution of the data is quite good. One of the unconformities was produced because of erosion in the geological time scale change at the depth of 2.5 km. These types of structures make imaging more complex; for example, below the fracture the interval velocity and migration of data might be wrong. Using the hybrid travel time method provides accurate velocity determination for migration. The target of our imaging was the reservoir, which at

2.5 km has an anticlinal structure. This was imaged properly, as shown in the interpreted section in Figure 12b.

The amplitude spectrum of the data is generally used to predict reservoir productivity by seismic imaging, as shown in Figure 13a. his approach uses the data in the time domain, while the output is the amplitude spectrum using a Fourier transform. Furthermore, a frequency spectrum is plotted against the three data sets for quantitative evaluation: (a) input shot gather data; (b) diffraction data; (c) migrated data (Figure 13b). This demonstrates a low-frequency enhancement with higher amplitude (blue), which is very important for imaging. A targeted zone below 2.5 km is resolved and can be interpreted for structural clarifications and for further reservoir characterization.

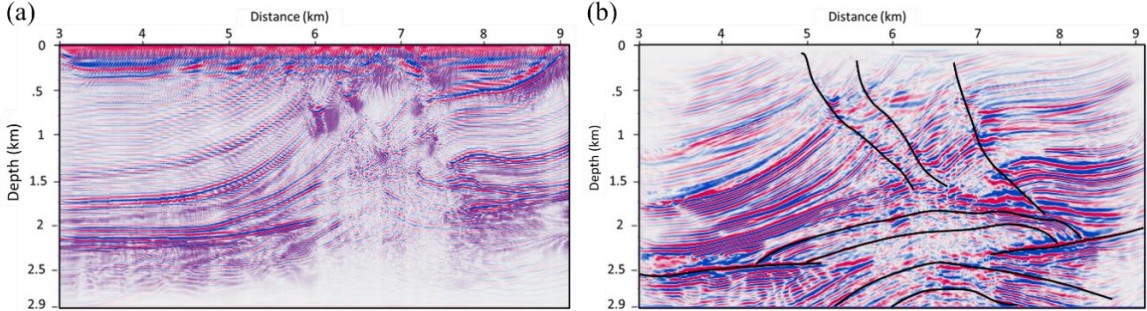

**Figure 12.** (**a**) Seismic migration using prestack Kirchhoff migration. A velocity model was updated using semblance velocity analysis. Faults and anticlinal structures that were our targets are not imaged. (**b**) Migrated seismic data using Kirchhoff depth migration, with the hybrid travel time calculated by the eikonal equation and paraxial raytracing. Faults and anticline sections in the deeper section are imaged properly.

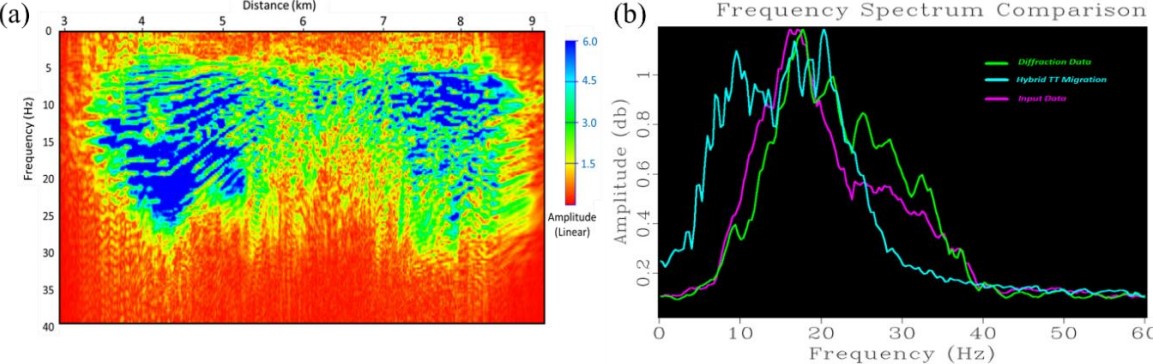

**Figure 13.** (**a**) The amplitude spectrum of the migrated data shows a linear distribution of the amplitude energy. (**b**) Frequency spectrum of input data (purple), diffraction data (green), and migrated data using hybrid travel time (blue).

In addition to providing high-resolution seismic data and a better signal-to-noise ratio, velocity information about the subsurface provide critical results. For this reason, an extensive velocity analysis was performed to select and prove the best method of velocity estimation. The hybrid travel time equation can successfully assess velocity, especially in the shadow zone and multipathing areas, which was proven by the results.

## 4. Conclusions

This paper shows the importance and behavior of seismic diffraction with respect to velocity and depth, which follows the Zoeppritz equation by considering the angle of incidence and not the offset. We found that three parameters have essential effects on diffraction behavior, namely the velocity,

depth, and frequency of the seismic wave. Further, the need for a diffraction separation method was studied and a DFF filtering technique was introduced for diffraction separation and imaging.

Finally, two methods of velocity model-building were used for the Marmousi dataset to demonstrate the importance of velocity modeling. The hybrid travel time method for calculating the velocity model was more accurate than the semblance method because it uses both an eikonal solver and paraxial raytracing to calculate the velocity in the shadow zones, such as for fractures. Use of the correct velocity model is the key to seismic depth imaging. The model enhanced the preserved energy through the recovery of low frequencies in the final image and helped us to interpret the fractures and deeper events that contribute to reservoir delineation in oil and gas exploration.

**Author Contributions:** Y.B. developed the algorithms, performed the research, and obtained the results. N.M.M. performed the geological studies on the area. S.Y.M.A. reviewed the final results and technically reviewed the geological aspects of the area. S.H.A. performed the regional geology of the study area. D.P.G. technically reviewed the work and helped to improve the results. All authors have read and agreed to the published version of the manuscript.

**Funding:** This research received no external funding.

**Acknowledgments:** We are thankful to Universiti Sains Malaysia (USM) and Universiti Teknologi PETRONAS (UTP) for providing the facility used for this research. This work is supported by the USM short-term research Grant. In this work, we used MATLAB and Seismic Unix from Colorado School of Mines (CSM).

**Conflicts of Interest:** The authors declare no conflict of interest.

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
