# Peer review of "Inspiration for Seismic Diffraction Modelling, Separation, and Velocity in Depth Imaging"

_applsci, doi:10.3390/app10124391_

Round 1
Reviewer 1 Report
This paper presents a topic worth investigation. The author has pretty good work on an experimental approach. But there are some issues have to be addressed before accepting:
- The literature review should be organized in a better format; I would suggest you use table summarizes findings from the literature review
- Some implication or data support from theory and field test is highly recommended, but not necessary, but at least discuss it in the paper.
- The future plan or suggestion is highly recommended.
- More related literature should be cited and discussed. The list of papers is shown below.
- Li, J., Chen, X., Hao, Z., and Rui, Z., 2005, A Study on multiple time-lapse seismic AVO inversion, Chinese Journal of Geophysics, 48(4), pp. 902-908
- Wang, Z., Bai, Y., Zhang, H., Liu, Y., 2019. Investigation on gelation nucleation kinetics of waxy crude oil emulsions by their thermal behavior. Journal of Petroleum Science and Engineering, doi: https://doi.org/10.1016/j.petrol.2019.106230
Author Response
Response to Reviewer 1 Comments
Dear Reviewer 1, Thank you so much for the reviewer paper and provide constructive comments which help to improve the article. We really appreciate your efforts and the time given to our work. I have replied the all the points and included in the paper which can be tracked in “Track change” paper and a clean version is also attached with the submission.
Point 1: The literature review should be organized in a better format; I would suggest you use table summarizes findings from the literature review
Response 1: I have organized and added more literature in chronological order in the statement as below.
(Further, the diffraction response for a nonzero separation of source and receiver is presented by Barryhill in 1977 [9] in which a theoretical support was obtained for applying the zero-separation theory to stack the seismic data, then the amplitude preservation of seismic wave is achieved by Hilterman in 1975 [10] by using the front sweep velicity approach in which a graphical method evolves. Last decade was a theoretical world and many methods was introduced but the lack of implementation in a real world as because of computing power and facilities. In this decade, the computation increases drastically, and the technology development increased compare to the last decade. The diffraction enhancement in prestack seismic data was introduced by Bansal in 2005 [11] in which he introduce the most effective techniques which is decomposition of seismic gather into eigensection and flows based on radon transformations. Diffraction imaging by multi-focusing method was used in 2009 by Berkovitch [12] in which the diffraction focusing stack (DMFS) used to separate the diffraction energy in a stack section and defocusing the reflection energy over a large area. Separating diffraction is widely used by Fomel since 2002 [13] using Claerbout principle [14] and has contribute a lot in the field of seismic diffraction imaging.)
Point 2: Some implication or data support from theory and field test is highly recommended, but not necessary, but at least discuss it in the paper.
Response 2: Implication and data examples published from theory in last decade and recently are discussed in the literature section (introduction), as the figures contain copyright issues and data is proprietary from the publisher. Because of that example data could not be shown in the paper.
Point 3: The future plan or suggestion is highly recommended.
Response 3: PA section is added in the last as title future plan and suggestions.
Future and Suggestion: The future of the seismic industry is finding small scale reservoir as large hydrocarbon traps are explored, the diffraction imaging provides the facility to explore these reservoirs which could bring the basin production up to higher rate. Further, the diffraction separation approach can be extended from frequency domain to time domain using plane-wave destruction and improve the filtering techniques for more accurately preserving diffraction for high resolution images.
Point 4: More related literature should be cited and discussed. The list of papers is shown below.
- Li, J., Chen, X., Hao, Z., and Rui, Z., 2005, A Study on multiple time-lapse seismic AVO inversion, Chinese Journal of Geophysics, 48(4), pp. 902-908
- Wang, Z., Bai, Y., Zhang, H., Liu, Y., 2019. Investigation on gelation nucleation kinetics of waxy crude oil emulsions by their thermal behavior. Journal of Petroleum Science and Engineering, doi: https://doi.org/10.1016/j.petrol.2019.106230
Response 4: 7 more references are added to the literature and one of the paper suggested was cited as the amplitude preservation is necessary for Amplitude Versus Offset (AVO) analysis in 4D seismic data processing and imaging.
[9] J. R. Berryhill, “DIFFRACTION RESPONSE FOR NONZERO SEPARATION OF SOURCE AND RECEIVER,” GEOPHYSICS, vol. 42, no. 6, pp. 1158–1176, 1977, doi: doi:10.1190/1.1440781.
[10] F. Hilterman, “Amplitudes of seismic waves; a quick look,” GEOPHYSICS, vol. 40, no. 5, pp. 745–762, 1975, doi: 10.1190/1.1440565.
[11] R. Bansal and M. G. Imhof, “Diffraction enhancement in prestack seismic data,” GEOPHYSICS, vol. 70, no. 3, pp. V73–V79, 2005, doi: 10.1190/1.1926577.
[12] A. Berkovitch, I. Belfer, Y. Hassin, and E. Landa, “Diffraction imaging by multifocusing,” Geophysics, vol. 74, no. 6, pp. WCA75–WCA81, 2009.
[13] S. Fomel, “Applications of plane-wave destruction filters,” Geophysics, vol. 67, no. 6, pp. 1946–1960, 2002.
[14] J. F. Claerbout, Earth soundings analysis: Processing versus inversion, vol. 6. Blackwell Scientific Publications Cambridge, Massachusetts, USA, 1992.
[20] J. LI, X. CHEN, Z. HAO, and Z. RUI, “A study on multiple time‐lapse seismic AVO inversion,” Chinese J. Geophys., vol. 48, no. 4, pp. 974–981, 2005.

Reviewer 2 Report
The paper deals with seismic diffraction in relation to seicmic imaging problems. The issue of diffraction separation methods and the effect of the velocity analysis method is considered. A number of examples of demonstration of the techniques under consideration are given. The work undoubtedly deserves attention, but there are a number of comments:
- Such notions as "fractures" and "faults" constantly appear in the work. I would like more specifics in the description of these objects and what they represent in the model that the authors use.
- In section 3.1, authors repeatedly emphasize that they use a fractured model. However, a much simpler, fracture-free model is used in the simulation. Why is there such an emphasis on fractures when they are not presented in any way?
- In section 3.1, the authors constructed a simplified model based on the real one. It contains faults. In this case, most likely, the fault is characterized by a subvertical or vertical interface between the media (in contrast to the geological layers). Which confirms the model chosen by the authors. Earlier on line 167, the authors write "... the wave response of small subsurface discontinuity, such as faults, ...", but in their model the fault is characterized by a subvertical boundary rather than a small subsurface discontinuity. This issue needs clarification.
- On lines 198-199, the authors write: "The focus of this research is to image the fractures which are the main challenges faced in the
field including fractured basement, fracture distribution ... ", but there are no fractured models in the calculations. Why then should this be the focus of the article? - In paragraph 3.4, the authors give a general picture of the fields and say that "Faults ... are resolved." This should be shown in more detail in the resulting images.
- In conclusion, the ators write: "... helps us to interpret the fractures corridors that contribute to fractured reservoir delineation in Oil & Gas exploration." (lines 431-432), but examples of fractured models are not considered in the work, an explanation of what is meant by this is required.
- General remark. The authors write about many geological heterogeneities, such as faults, fractures, channels, rough edges of structures and karst. In fact, the authors show the possibility of distinguishing subvertical interfaces between surfaces during migration. An explanation of the conclusions about the applicability of these approaches for the considered geological objects is required.
Author Response
Response to Reviewer 2 Comments
Dear Reviewer 2, Thank you so much for the reviewer paper and provide constructive comments which help to improve the article. We really appreciate your efforts and the time given to our work. I have replied the all the points and included in the paper which can be tracked in “Track change” paper file and a clean version is also attached with the final submission.
Point 1: Such notions as "fractures" and "faults" constantly appear in the work. I would like more specifics in the description of these objects and what they represent in the model that the authors use.
Response 1: In general Geology, the fracture is any kind of separation or break in a rock formation. Examples are joints or faults which divide the formation into two or more pieces. The fracture itself is a broad term that includes in Geology the faults, fracture, and discontinuities. These fractures can provide access for fluids, like water or hydrocarbons to move into the rocks. In the model, we used the general term “Fracture” which includes the fault and acoustic impedance contract in a velocity model.
Point 2: In section 3.1, authors repeatedly emphasize that they use a fractured model. However, a much simpler, fracture-free model is used in the simulation. Why is there such an emphasis on fractures when they are not presented in any way?
Response 2: As explained in point 1 that fracture is a broad term that is used for fault and joint, no matter is a small discontinuity or a large impedance contrast which case of formation drop down because of tectonic activities. Showing a model is to explain the fault geometer and production of diffraction hyperbola accurately.
Point 3: In section 3.1, the authors constructed a simplified model based on the real one. It contains faults. In this case, most likely, the fault is characterized by a subvertical or vertical interface between the media (in contrast to the geological layers). Which confirms the model chosen by the authors. Earlier on line 167, the authors write "... the wave response of small subsurface discontinuity, such as faults, ...", but in their model the fault is characterized by a subvertical boundary rather than a small subsurface discontinuity. This issue needs clarification.
Response 3: We have used two models, a fractured model from the Malay basin and the other is the Marmousi velocity model. In line 167, we refer to the small-scale discontinuities in the model where three major faults together with small changes in velocity contrast appear at the bottom where a pinchout appears because of unconformity. So, basically, the fracture word itself is explaining faults and fractures either large scale or small scale. I have added the explanation of fractures in the paper.
Point 4: On lines 198-199, the authors write: "The focus of this research is to image the fractures which are the main challenges faced in the
field including fractured basement, fracture distribution ... ", but there are no fractured models in the calculations. Why then should this be the focus of the article?
Response 4: “The focus of this research is to image the fractures which are the main challenges faced in the field including fractured basement, fracture distribution, connectivity, and lateral variation, which caused poor seismic imaging.” is the full sentence, in which we are focusing the faults which produce because of lateral changes in velocity either on a large scale or small scale. And diffraction production is certain in any case either small or large discontinuity. So, the focus of research remains to image these types of structures produced by AI contracts either faults, fractures, channels, rough edges of structures, and karst. No matter the structure type, but the response is important which is captured and preserved in the data to image.
Point 5: In paragraph 3.4, the authors give a general picture of the fields and say that "Faults ... are resolved." This should be shown in more detail in the resulting images.
Response 5: Explanation of the figure is added together with the interpreted section of Faults and anticline structure. “Further, the three major faults are interpreted accurately as the resolution of data is quite good. One of the unconformities, which was produced because of erosion in geological time scale change at the depth of 2.5 km. These types of structures make imaging more complex such as below fracture, the interval velocity identical and migration of data might be wrong. But, here using a hybrid travel time method provides the accurate velocity determination for migration. As the target of our imaging was the reservoir which at 2.5 km anticlinal structure is imaged properly as shown in interpreted section Figure 12b.”
Point 6: In conclusion, the authors write: "... helps us to interpret the fractures corridors that contribute to fractured reservoir delineation in Oil & Gas exploration." (lines 431-432), but examples of fractured models are not considered in the work, an explanation of what is meant by this is required.
Response 6: As explained above the fracture corridor mean multiple faults which are in connection to transfer the oil and gas to the reservoirs, also in the case of fractured basement these fractures are interconnected to increase the porosity and permeability of the reservoir. The imaging of these fractures are similar to what we are doing in these two model data. The important thing is to prove the concept on a simple model and develop the algorithm which would be later extended to the real fractured reservoir as mentioned in the future plan and suggestions.
Point 7: General remark. The authors write about many geological heterogeneities, such as faults, fractures, channels, rough edges of structures, and karst. In fact, the authors show the possibility of distinguishing subvertical interfaces between surfaces during migration. An explanation of the conclusions about the applicability of these approaches for the considered geological objects is required.
Response 7: We have written the general introduction about using diffraction importance in case of these geological heterogeneities such as faults, fracture, channels, rough edges of the salt body, and karst. In these cases, the diffraction imaging is an identical process which increases the resolution of seismic data with higher accuracy of interpretation such features. These types of structure are not easy to get in the synthetic data because of that we have chosen the model which are similar and provides the authenticity of real environment of at least faults and fractures.
Further, the methodology section is described separately in “ Generalized Workflow” to explain the methodology in detail and step by step in 7 points to understand the research work.
Also, the addition to explaining the results been added for final results, and the last results are interpreted to explain and understand the authenticity of the proposed method in better imaging such as faults, anticlines, pinchouts, unconformity, and deeper imaging below 2.5 km. which proves the method chosen for research is adequately applicable for high-resolution imaging.

Round 2
Reviewer 2 Report
Now the work can be published.